# Frontotemporal dementia mutant Tau promotes aberrant Fyn nanoclustering in hippocampal dendritic spines

Pranesh Padmanabhan[†], Ramón Martínez-Mármol[†], Di Xia, Jürgen Götz*, Frédéric A Meunier*

Clem Jones Centre for Ageing Dementia Research (CJCADR), Queensland Brain Institute (QBI), University of Queensland, Brisbane, Australia

**Abstract** The Src kinase Fyn plays critical roles in memory formation and Alzheimer's disease. Its targeting to neuronal dendrites is regulated by Tau via an unknown mechanism. As nanoclustering is essential for efficient signaling, we used single-molecule tracking to characterize the nanoscale distribution of Fyn in mouse hippocampal neurons, and manipulated the expression of Tau to test whether it controls Fyn nanoscale organization. We found that dendritic Fyn exhibits at least three distinct motion states, two of them associated with nanodomains. Fyn mobility decreases in dendrites during neuronal maturation, suggesting a dynamic synaptic reorganization. Removing Tau increases Fyn mobility in dendritic shafts, an effect that is rescued by re-expressing wildtype Tau. By contrast, expression of frontotemporal dementia P301L mutant Tau immobilizes Fyn in dendritic spines, affecting its motion state distribution and nanoclustering. Tau therefore controls the nanoscale organization of Fyn in dendrites, with the pathological Tau P301L mutation potentially contributing to synaptic dysfunction by promoting aberrant Fyn nanoclustering in spines.
DOI: https://doi.org/10.7554/eLife.45040.001

*For correspondence:
j.goetz@uq.edu.au (JG);
f.meunier@uq.edu.au (FAM)

[†]These authors contributed equally to this work

**Competing interests:** The authors declare that no competing interests exist.

## Introduction

Dendritic spines compartmentalize biochemical reactions that are critical for synaptic plasticity, which underpins memory and learning. A myriad of signaling molecules acts in a spatiotemporally controlled manner to translate the information encoded in post-synaptic calcium influx into appropriate changes in synaptic strength during learning (*Nishiyama and Yasuda, 2015*). A central signaling role in the spine is assumed by the tyrosine kinase Fyn, a member of the Src family, which is widely expressed throughout the brain (*Ohnishi et al., 2011*). Fyn is myristoylated, a process that occurs co-translationally on free ribosomes, and is subsequently palmitoylated, which enhances the hydrophobicity of the molecule and membrane association (*Sato et al., 2009*). The synaptic scaffolding protein PSD-95 is also palmitoylated and known to bind membrane-associated Fyn, recruiting it into the proximity of the GluN2B subunit of the NMDA receptor (NMDAR), thereby enhancing Fyn-mediated phosphorylation of GluN2B (*Tezuka et al., 1999*). This results in an increased stability of the NMDAR complex at the synaptic membrane, which is critical for synaptic plasticity (*Prybylowski et al., 2005*).

Fyn acts as a molecular hub that interacts with multiple synaptic proteins and controls major signaling pathways (*Nishiyama and Yasuda, 2015*). A functional role for Fyn in the dendritic compartment is underscored by the finding that Fyn knockout mice exhibit a decreased spine density in layer V pyramidal neurons (*Morita et al., 2006*), reduced axonal branching in the cerebellar cortex (*Cioni et al., 2013*), and deficits in long-term potentiation and spatial learning (*Grant et al., 1992*). Fyn requires the microtubule-associated protein Tau, a protein that is implicated in

neurodegenerative diseases including Alzheimer's disease (AD) and frontotemporal dementia (FTD), for its efficient targeting to the dendritic compartment (*Ittner et al., 2010*; *Xia et al., 2015*). This process is facilitated by the interactions of Tau with the SH3 domain of Fyn and of tyrosine-phosphorylated Tau with the SH2 domain of Fyn (*Lee et al., 1998*; *Bhaskar et al., 2005*). Further, increased Tau expression (in particular that of mutant Tau forms found in FTD, such as P301L Tau) is associated with increased synaptic localization, not only of Tau but also of Fyn (*Ittner et al., 2010*; *Xia et al., 2015*; *Hoover et al., 2010*).

Fyn acts as a key mediator of two central molecules in AD, amyloid-β (Aβ) and Tau, which forms amyloid plaques and neurofibrillary tangles, respectively, with Aβ lying upstream of Tau in the pathocascade (*Götz et al., 2001*). Aβ acts on Fyn via multiple receptors that are located on the plasma membrane, including the prion protein PrP$^c$ (*Um et al., 2012*; *Larson et al., 2012*; *Um et al., 2013*). Activated Fyn phosphorylates NMDARs and mediates interactions between NMDAR and PSD-95, which are required for Aβ-induced excitotoxicity (*Um et al., 2012*). Fyn further mediates Aβ-induced local protein translation and the accumulation of Tau in the somatodendritic compartment by activating the ERK/rpS6 signaling pathway (*Li and Götz, 2017*). Fyn overexpression also accelerates cognitive impairment (*Chin et al., 2005*; *Kaufman et al., 2015*), whereas depleting Fyn or inhibiting its activity restores memory function and synaptic density in AD model mice (*Chin et al., 2004*). Therefore, both Aβ toxicity and Tau pathology involve Fyn kinase activity (*Ittner and Götz, 2011*; *Haass and Mandelkow, 2010*).

How Fyn integrates these diverse signals in subcellular compartments such as spines is currently unknown. It is difficult to conceive how individual Fyn molecules can act as a nexus that is capable of integrating such a variety of signals. Whether Fyn is spatiotemporally organized to mediate efficient signal transduction remains to be established. The spatial organization of receptors and signaling molecules into nanodomains in biological membranes is emerging as an essential feature of cell signaling (*Kusumi et al., 2012*). These nanodomains are formed by a combination of protein–protein, lipid–lipid, protein–lipid and cytoskeletal interactions (*Milovanovic and Jahn, 2015*; *Goyette and Gaus, 2017*; *Padmanabhan et al., 2019*), as well as by membrane-mediated forces (*Johannes et al., 2018*). Consequently, these nanodomains concentrate various molecules in discrete areas, thereby facilitating efficient and robust processing of cellular information by regulating a complex series of biochemical reactions (*Harding and Hancock, 2008*). Indeed, a recent study has demonstrated that ligand-induced CD36 receptor clustering promotes Fyn activation within these clusters in non-neuronal cells (*Githaka et al., 2016*). Whether Fyn concentrates in such nanodomains in the dendrites, and whether Tau regulates the nanoscale organization of Fyn, is not known but such mechanisms could underlie the pleiotropic roles that Fyn assumes in neuronal signaling.

The advent of super-resolution microscopy has paved the way for investigations of the nanoscale organization and dynamic behavior of receptors (*Nair et al., 2013*; *Hoze et al., 2012*), as well as for studies on their signaling (*Lu et al., 2014*), trafficking (*Joensuu et al., 2016*) and scaffolding molecules (*Chamma et al., 2016*). Here, we used single-particle tracking photoactivated localization microscopy (sptPALM) to determine whether Tau controls the organization of Fyn in the somatodendritic compartment. We found that dendritic Fyn displayed a nanocluster organization that is underpinned by multiple mobility states, and that Fyn mobility significantly decreased in dendrites with neuronal maturation. In neurons from Tau knockout (Tau KO) mice, Fyn mobility increased in the dendritic shafts, an effect that was rescued by the re-expression of wildtype (WT) Tau. More importantly, pathological P301L mutant Tau, as found in familial FTD, but not a truncated form of Tau lacking the microtubule-binding domain (ΔTau), promoted the trapping of Fyn in the dendritic spines. Our study therefore reveals a complex interplay between Fyn and Tau in the somatodendritic compartment and points to a novel role of altered Fyn nanoclustering in causing synaptic dysfunction in disease.

## Results

### sptPALM reveals changes in the dendritic nanoscale organization of Fyn as neurons mature

To investigate the spatial distribution and mobility pattern of the kinase Fyn in live neurons, we fused Fyn with the photoconvertible fluorescent protein mEos2 (Fyn-mEos2), and expressed

this fusion protein in hippocampal neurons obtained from WT mice. We then performed sptPALM, using mCardinal as a cytoplasmic marker (*Figure 1a,b*). We used an oblique illumination configuration to image the dendrites of neurons at a stage when they predominantly exhibited filopodia (11–15 days in vitro (DIV)) and when they had developed mature spines (DIV19-20) (*Figure 1a–d*). By applying a weak intensity 405 nm laser, Fyn-mEos2 molecules were randomly photoconverted from a green- to a red-emitting state at a low spatial density, such that individual Fyn-mEos2 molecules could be correctly localized and tracked. The photoconverted molecules were then detected in the red-emitting channel at 561 nm excitation at 50 Hz for a duration of 320 s (16,000 frames), allowing for the characterization of Fyn distribution in live neurons at a high spatiotemporal resolution.

The low-resolution epifluorescence image of Fyn-mEos2 suggested a relatively uniform distribution of Fyn in the dendrites, whereas the sptPALM localization density map, which was generated by binning all the localizations acquired over the 16,000 frames, revealed a very heterogeneous spatial distribution of localization densities in dendrites and even within individual spines (*Figure 1e,f*). Regions with high localization density tended to have a higher local concentration of Fyn molecules than regions with low localization density (*Cisse et al., 2013*), suggesting the presence of Fyn-enriched nanodomains in dendrites. Nanodomains generally stem from the lateral trapping of molecules in discrete areas of the plasma membrane. We therefore computed the trajectories of Fyn-mEos2 molecules lasting for at least eight frames and generated a spatial map of diffusion coefficients, with Fyn mobility values being directly proportional to the diffusion coefficients (*Figure 1g,h*). We found that Fyn mobility was spatially heterogeneous and that the diffusion coefficients of Fyn-mEos2 molecules varied more than 100-fold in individual neurons and even within individual spines (*Figure 1—figure supplement 1*). Using a Voronoï-tessellation-based spatial clustering algorithm (*Levet et al., 2015*), we identified potential Fyn nanodomains (or nanoclusters) within spines and estimated the diameter of these nanodomains to be ~168.3 ± 4.2 nm (mean ± SEM; n = 502 nanodomains; *Figure 1—figure supplement 2*). This is comparable to the size of PSD-95 nanodomains in spines measured using sptPALM (*Nair et al., 2013*) and is around three times smaller than the size of a spine head (*Izeddin et al., 2012*). These results establish the existence of discrete Fyn nanodomains in the somatodendritic compartment, and highlight the capability of sptPALM in examining single-molecule and population-level dynamics of Fyn molecules in individual cells at high resolution.

Given that Fyn has a critical role in integrating signaling pathways in the dendritic compartment (*Li and Götz, 2017*), we next investigated whether the mobility of Fyn changes during maturation by analyzing filopodia-forming DIV11-15 neurons and spine-forming DIV19-20 neurons. First, we computed the mean square displacement (MSD) of individual trajectories lasting for at least eight frames, and then calculated the average of the MSDs of all trajectories from each analyzed cell (*Figure 1k*). We used the area under the average MSD curve (AUC) of each cell for statistical comparisons (*Figure 1l*). We then estimated the diffusion coefficients from the MSDs of each trajectory and generated a frequency distribution of the diffusion coefficients of all trajectories from each analyzed neuron (*Figure 1m*). The distributions of the diffusion coefficients were then grouped into two fractions, immobile and mobile, on the basis of a threshold value of the diffusion coefficient, as described previously (*Constals et al., 2015*) (*Figure 1—figure supplement 3*), with the immobile fraction being used for statistical comparisons (*Figure 1n*). We detected Fyn-mEos2 molecules in both the spines and shaft regions at both time points of maturation (*Figure 1e–j*), and noted a frequent exchange of Fyn-mEos2 molecules between these two compartments. Interestingly, we found that the average MSD and AUC decreased and the immobile fraction increased significantly with dendritic maturation, and that the frequency distribution of diffusion coefficients shifted to the left with dendritic maturation, demonstrating that Fyn mobility decreased as dendrites matured. These observations suggest that the extent of lateral trapping of Fyn in dendrites increases with neuronal development and spine maturation, possibly reflecting the dynamic changes in the composition of Fyn-binding partners in dendrites.

## Fyn mobility is lower in spines than in dendritic shafts

Fyn has multiple substrates in the dendritic compartment, including Tau (*Ittner et al., 2010*), PSD-95 (*Won et al., 2016*), and Pyk2 (*Li and Götz, 2018*). These proteins are also binding partners of Fyn, with their abundance and composition likely to be spatially regulated in the shafts and spines. For instance, it has been reported that PSD-95 is enriched in spines, whereas Tau levels in this compartment are low unless the protein either carries a pathological mutation found in familial cases of FTD

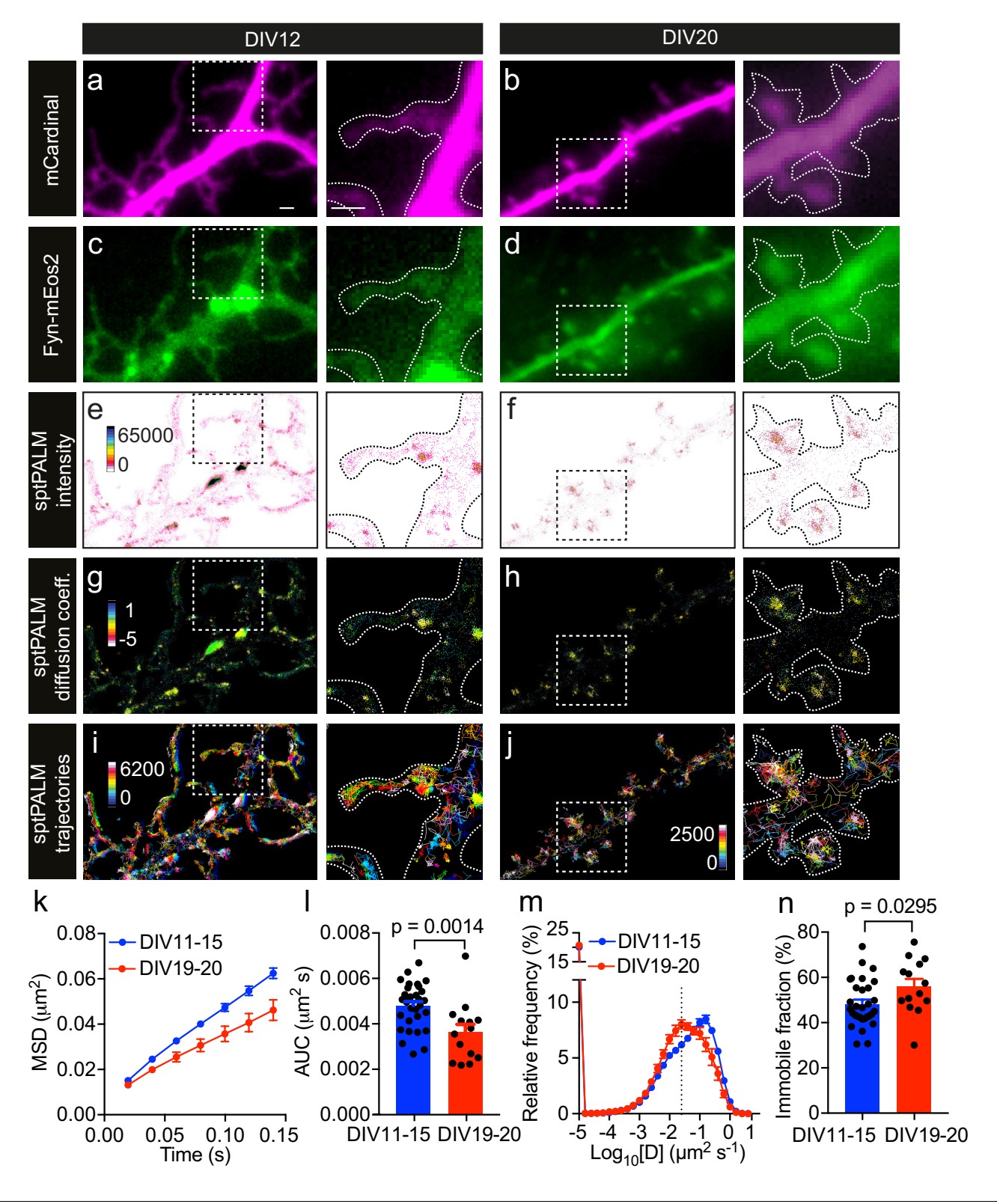

**Figure 1.** Fyn mobility decreases with dendritic spine maturation. The analysis was performed at days in vitro (DIV)11–15 and DIV19–20. (a–d) Representative epifluorescence images of DIV12 (a, c) and DIV20 (b, d) neurons co-expressing mCardinal (a, b) and Fyn-mEos2 (c, d), acquired before photoconversion of mEos2 molecules. Insets are shown at a higher magnification. Scale bars, 1 μm. (e–j) sptPALM imaging was performed at 50 Hz for 320 s (16,000 frames) to construct the maps of the localization intensities (e, f), diffusion coefficients (g, h) and trajectories (i, j) of Fyn-mEos2 molecules.

*Figure 1 continued on next page*

*Figure 1 continued*

The cooler colors represent higher localization intensities (e, f) and larger diffusion coefficients (g, h), and each trajectory is coded with a different color (i, j). (k–n) Comparison of Fyn mobility parameters with development. (k) Average mean square displacement (MSD) as a function of time. (l) The corresponding area under the MSD curves (AUC). (m) Distribution of diffusion coefficients (D) shown in a semi-log plot. The threshold to distinguish the immobile ($Log_{10}[D] \leq -1.6$) and the mobile ($Log_{10}[D] > -1.6$) fraction of molecules is indicated with a dashed line. (n) The corresponding immobile fraction. Error bars are standard errors of the mean (SEM). Mean ± SEM values were obtained from n = 31 neurons (DIV11–15) and n = 14 neurons (DIV19–20). Statistical comparisons were performed using the Mann-Whitney U test (l) and Student's *t*-test (n).

DOI: https://doi.org/10.7554/eLife.45040.002

The following figure supplements are available for figure 1:

**Figure supplement 1.** Fyn mobility is heterogeneous even within individual spines.
DOI: https://doi.org/10.7554/eLife.45040.003

**Figure supplement 2.** The Voronoï-tessellation-based spatial clustering algorithm identifies Fyn nanodomains in dendritic spines.
DOI: https://doi.org/10.7554/eLife.45040.004

**Figure supplement 3.** Mean square displacement analysis.
DOI: https://doi.org/10.7554/eLife.45040.005

or is hyperphosphorylated (*Hoover et al., 2010*). To investigate whether Fyn mobility differs between the dendritic shafts and spines of hippocampal neurons (DIV20-22), we co-expressed Fyn-mEos2 and enhanced green fluorescence protein (EGFP) as a volume marker to identify each compartment, and compared the mobility of Fyn-mEos2 in dendritic shafts and spines. We first acquired image stacks of the epifluorescence GFP signal, which allowed us to create a three-dimensional (3D) reconstruction of the dendritic structure (*Figure 2a–c*). We then performed sptPALM of Fyn-mEos2. Only spines protruding from the sides of the dendrite branch were considered in our analysis (*Figure 2a–c*), with trajectories being extracted from these spines (color-coded in blue in *Figure 2d*). We next identified trajectories from the dendritic shaft region (color-coded in red in *Figure 2d*) by excluding the shaft segments containing spines projecting away from the imaging plane (white arrows in *Figure 2b,c* and black arrows in *Figure 2d*). This procedure allowed us to compare Fyn mobility in the dendritic shafts and spines for every analyzed cell. Remarkably, we found that the mobility of Fyn-mEos2 was significantly lower in the spines than in the shafts, as assessed by changes in the average MSD (*Figure 2e*), the AUC (*Figure 2f*), the distribution of diffusion coefficients (*Figure 2g*) and the immobile fraction (*Figure 2h*). These results suggest a higher abundance of binding partners or sites for Fyn in spines than in dendritic shafts.

## Fyn exhibits multiple kinetic subpopulations within shafts and spines

To further characterize Fyn mobility patterns in dendritic shafts and spines, we first performed a moment scaling spectrum (MSS) analysis of Fyn trajectories that lasted for at least 20 frames (0.4 s) and estimated the slope of the MSS ($S_{MSS}$). The MSS analysis revealed that Fyn-mEos2 molecules exhibit at least three different motion types in spines and shafts: immobile, confined and free diffusive states (*Figure 3a*). Trajectories with an $S_{MSS}$ close to 0 represent immobile molecules, those with an $S_{MSS}$ between 0 and 0.5 represent confined molecules, and those with an $S_{MSS}$ close to 0.5 represent apparently free molecules. We next analyzed the cumulative distribution of displacements of Fyn-mEos2 molecules at 20 ms intervals by applying a three-diffusive state model, with each state being differentiated by its diffusion coefficient. This model provided an excellent fit to the data (*Figure 3b*) and yielded estimates of apparent diffusion coefficients and state occupancies for each state, with the latter representing the fraction of Fyn molecules in each state (*Figure 3c, d*). We found that the apparent diffusion coefficients of the three diffusive states were similar in shafts and spines (*Figure 3c*), suggesting that similar molecular mechanisms could give rise to different mobility states of Fyn in both compartments. We associated the state with the lowest diffusion coefficient ($S_1$) to immobilized Fyn molecules, and the state with the intermediate diffusion coefficient ($S_2$) to confined Fyn molecules, both states constituting Fyn molecules within nanodomains. The state with the largest diffusion coefficient ($S_3$) represents freely moving Fyn molecules that are found outside nanodomains. Interestingly, the occupancies of the immobile and confined states were significantly higher and the occupancy of the free state significantly lower in the spines compared to shafts (*Figure 3d*), suggesting that the exchange between and the retention in different subpopulations of Fyn are differentially regulated in spines and shafts.

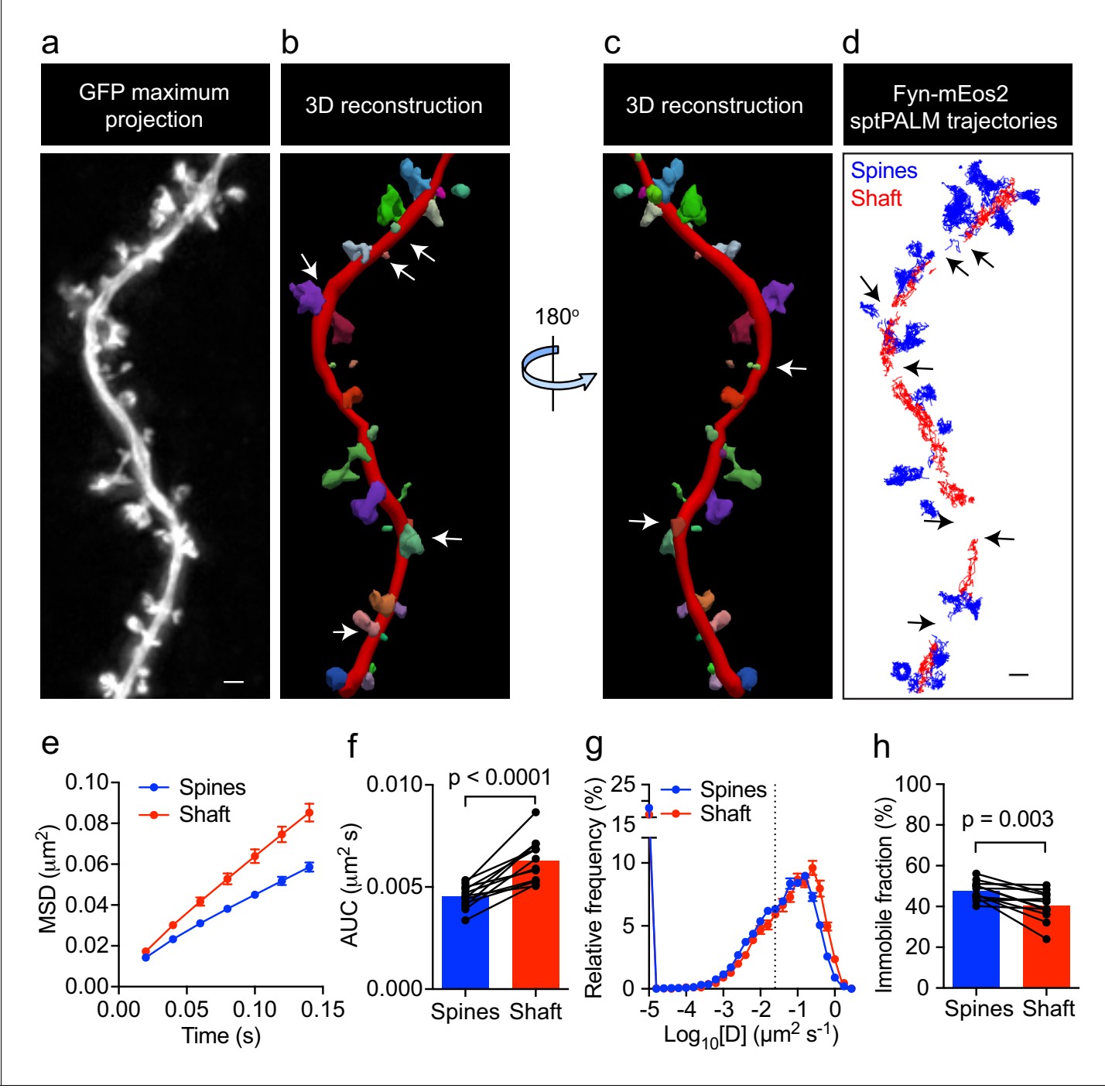

**Figure 2.** Fyn mobility is lower in spines than in shafts. (**a**) An EGFP image stack of a representative hippocampal neuron co-expressing Fyn-mEos2 and EGFP is shown as a two-dimensional (2D) maximum intensity projection along the z-direction. (**b**) 3D reconstruction of the same neuron using Neurolucida. (**c**) The image shown in (**b**) rotated 180˚ along the y-axis. (**d**) Trajectories belonging to spines that were nearly parallel to the sptPALM imaging plane (blue) and the dendritic shaft region (red) of the same neuron shown in (**a**). Dendritic segments containing spines projecting away from the sptPALM imaging plane were discarded (white arrows in (**b**) and (**c**), and black arrows in (**d**)). (**e–h**) Comparison of Fyn mobility parameters in dendritic shafts and spines. (**e**) Average mean square displacement (MSD) as a function of time. (**f**) The corresponding area under the MSD curves (AUC). (**g**) The distribution of diffusion coefficients (D) shown in a semi-log plot. The threshold used to distinguish the immobile ($Log_{10}[D] \leq -1.6$) and mobile ($Log_{10}[D] > -1.6$) fractions of molecules is indicated with a dashed line. (**h**) The corresponding immobile fractions. Error bars are SEM. Mean ± SEM values were obtained from n = 12 neurons. Statistical comparisons were performed using a paired Student's *t*-test (**f, h**).

DOI: https://doi.org/10.7554/eLife.45040.006

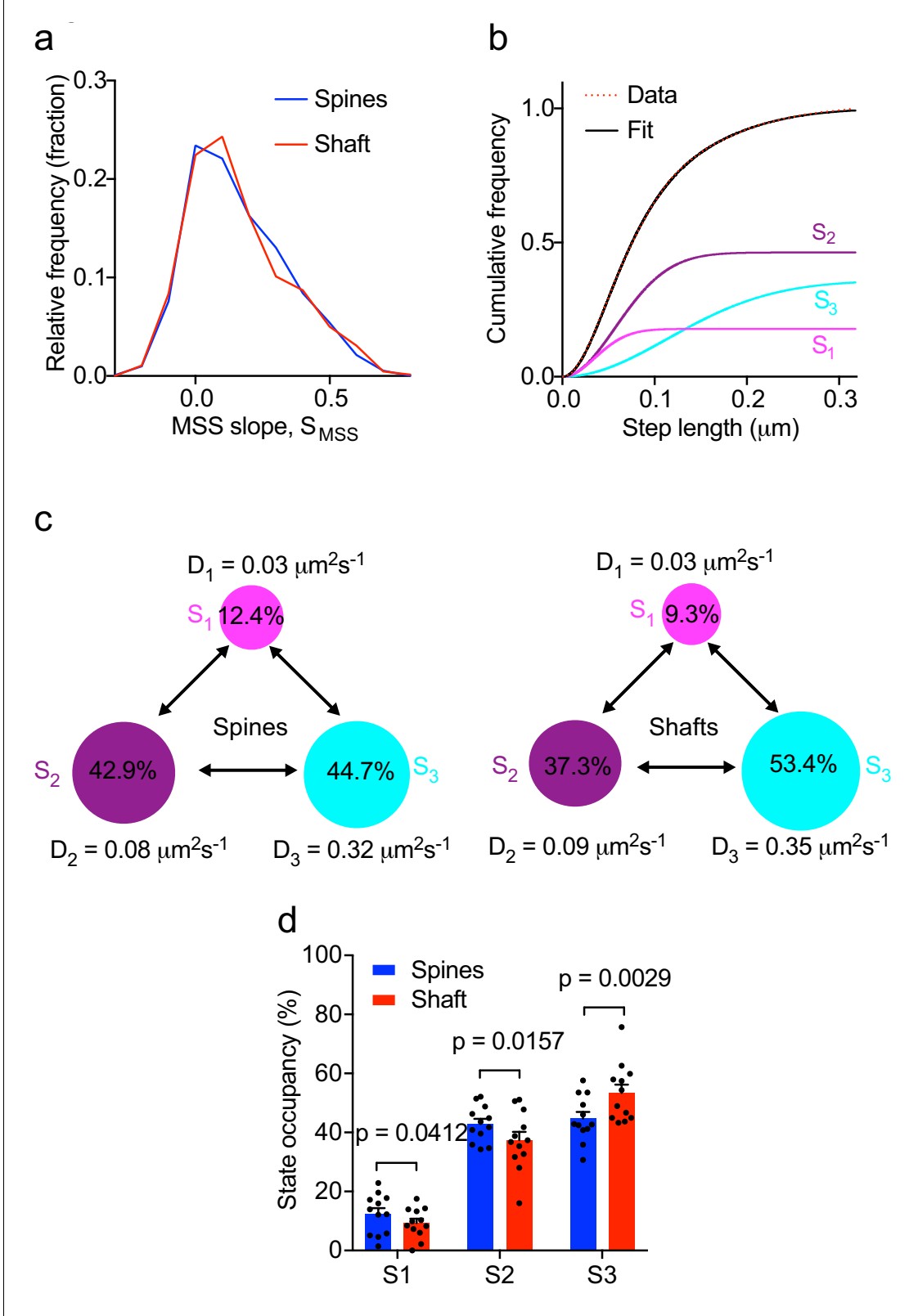

**Figure 3.** Multiple kinetic populations of Fyn in spines and shafts. (a) Distribution of the slope of the moment scaling spectrum ($S_{MSS}$) of trajectories lasting for at least 20 frames. (b) Representative fit (solid black line) of a three-diffusive-state model (*Equation (1)*) of the cumulative distribution of displacements at 20 ms intervals from Fyn-mEos2 molecules located inside the spines and shafts of neurons. The contribution of each diffusive state is shown individually. (c) A three-state model in which the estimated apparent diffusion coefficients and state occupancies are represented (state S1,

*Figure 3 continued on next page*

*Figure 3 continued*

immobile; state S2, confined; state S3, apparently free), with the circle area being proportional to the state occupancy. (d) Comparison of the estimated state occupancies of Fyn-mEos2 molecules in spines (blue) and shafts (red). Statistical comparisons were performed using paired Student's *t*-tests .
DOI: https://doi.org/10.7554/eLife.45040.007

## Tau controls the nanoscale organization of Fyn in hippocampal dendrites

Given that Tau binds directly to Fyn (*Lee et al., 1998*; *Bhaskar et al., 2005*), controls the dendritic targeting of Fyn (*Ittner et al., 2010*) and facilitates the recruitment of Fyn to the PSD-95 complex in spines (*Ittner et al., 2010*), we next investigated whether Tau also controls the nanoscale organization of Fyn in dendrites. Under physiological conditions, Tau is predominantly located in axons, with a smaller fraction of Tau being found in dendritic shafts, and an even smaller fraction in spines. We first confirmed our previous, biochemical finding (*Ittner et al., 2010*) that the dendritic targeting of Fyn is significantly reduced in Tau knock-out (Tau KO) compared to wildtype hippocampal neurons using confocal microscopy. Although the Fyn localization intensity was significantly decreased in the dendrites of Tau KO neurons compared to those of WT neurons, we were still able to observe Fyn localization in both the dendritic shafts and spines of Tau KO neurons, albeit at much lower levels (*Figure 4a–c*).

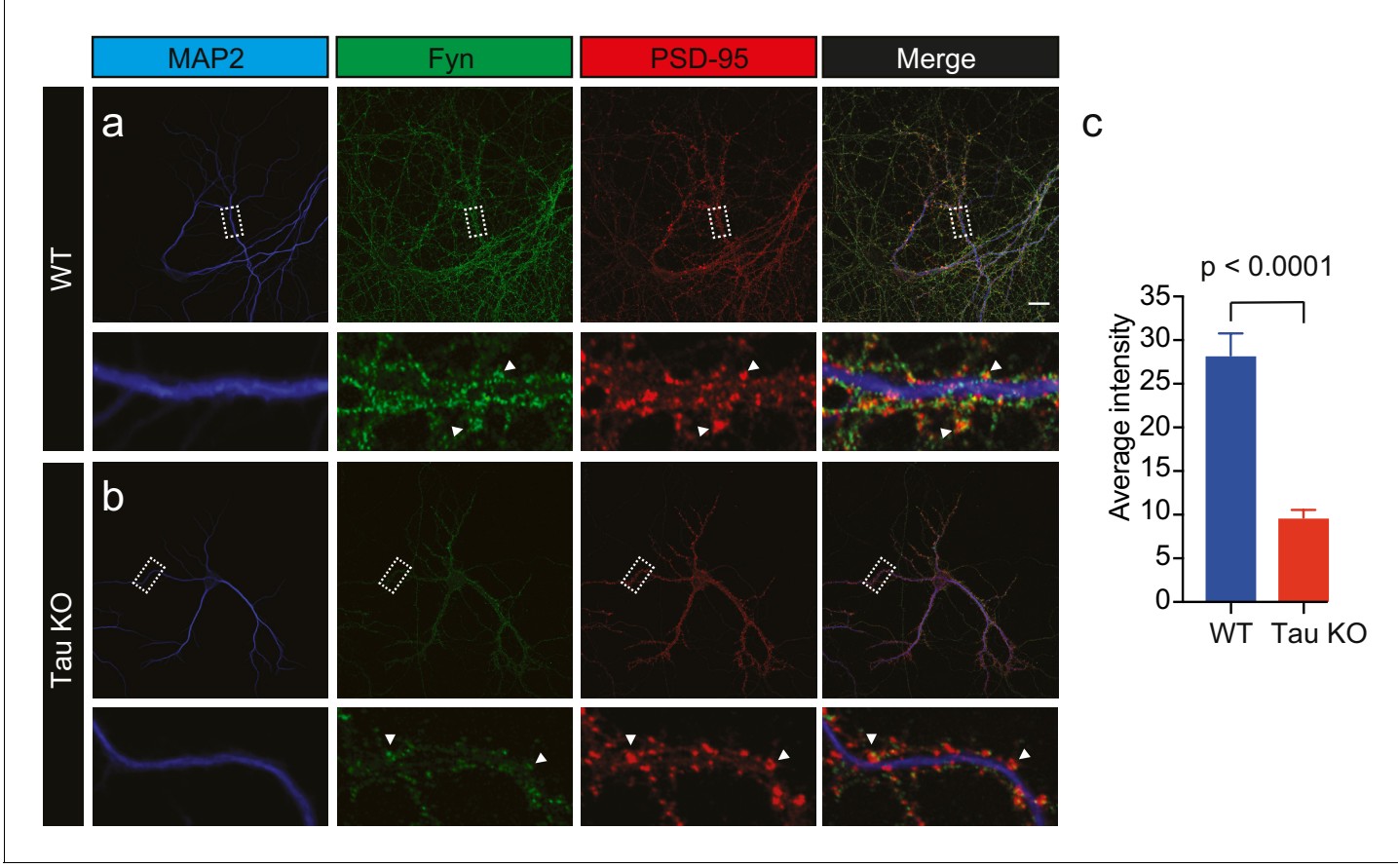

**Figure 4.** Loss of Tau reduces the localization of endogenous Fyn into the dendritic arbor. Immunocytochemistry of Fyn to characterize the distribution of endogenous Fyn along the dendritic arbor in wildtype (WT) and Tau knock-out (KO) mouse hippocampal neurons. (a, b) Panels showing the distribution of Fyn (green) in WT neurons (a) and Tau KO neurons (b). MAP2 (blue) was used to identify dendrites and PSD-95 (red) was used to identify dendritic spines. (c) Quantification of the average intensity of Fyn immunofluorescence in dendrites from WT and Tau KO neurons. Mean ± SEM values are calculated from n = 13 neurons. A statistical comparison of the average intensities was performed using a Student's *t*-test. Scale bars, 20 μm.
DOI: https://doi.org/10.7554/eLife.45040.008

We next performed sptPALM on WT and Tau KO neurons co-expressing Fyn-mEos2 and mCardinal at DIV19–22 and examined the effect of endogenous Tau expression on Fyn mobility (*Figure 5*). We detected Fyn-mEos2 molecules in both dendritic shafts and spines, and found that Fyn-mEos2 mobility was spatially heterogeneous in the dendrites of both WT and Tau KO neurons (*Figure 5a–d*). Although the mobility of Fyn-mEos2 appeared to be higher in the dendrites of Tau KO neurons compared to those of WT neurons, this difference was not statistically significant when the average MSD (*Figure 5e*), the AUC (*Figure 5f*), the frequency distribution of diffusion coefficients (*Figure 5g*) and the immobile fraction (*Figure 5h*) were determined. We then focused explicitly on Fyn in spines and determined whether its mobility was altered in Tau KO compared to WT neurons. This analysis revealed that the average MSD (*Figure 5i*) and the frequency distribution of diffusion coefficients (*Figure 5k*) were again remarkably similar and that the corresponding AUC (*Figure 5j*) and immobile fraction (*Figure 5l*) were not significantly different, indicating that the absence of Tau did not affect the nanoscale organization of Fyn in the spines. In other words, although Tau has a role in targeting Fyn to the dendritic compartment (*Ittner et al., 2010*), the nanoscale organization of the fraction of Fyn that entered this compartment was not affected by Tau knockout.

We then asked whether the absence of Tau affected Fyn mobility in the dendritic shafts. To investigate this, we identified shaft segments that were immediately adjacent to the spines and computed the effective diffusion coefficient ($D_{eff}$) of Fyn trajectories belonging to each spine and the corresponding shaft segment (*Figure 5m–o* and *Figure 5—figure supplement 1*). The $D_{eff}$ values for individual spines in WT and Tau KO neurons were similar (*Figure 5n*), whereas the $D_{eff}$ values for the shaft segments of Tau KO neurons were significantly higher than those of WT neurons (*Figure 5o*), suggesting that dendritic Tau regulates the nanoscale organization of Fyn in the shafts from which spines protrude. To determine whether the expression of Tau in Tau KO neurons can rescue this effect on Fyn mobility, we performed sptPALM of Tau KO neurons co-expressing Fyn-mEos2, mCardinal and TauWT-GFP (*Figure 5m–o*). As expected, the $D_{eff}$ values of Fyn in the shaft segments of Tau KO neurons expressing TauWT-GFP were not significantly different from those of WT neurons (*Figure 5o*).

Together, these results demonstrate that Tau regulates the mobility of Fyn in dendritic shafts, but not in spines, under physiological conditions. The increased trapping of Fyn in the shaft may contribute to the Tau-mediated dendritic targeting of Fyn.

## FTD-linked P301L mutant Tau traps Fyn in dendrites

Under disease conditions such as AD, Tau accumulates in spines. This has also been shown for Tau carrying the P301L mutation found in familial cases of FTD, which mislocalizes to dendritic spines (*Xia et al., 2015*; *Hoover et al., 2010*) and has an increased affinity towards Fyn in vitro (*Bhaskar et al., 2005*). Caspase-2 cleavage of Tau at asparagine 314 (Asp314) and the consequent formation of the truncation product ΔTau314 contributes to the mislocalization of pathological forms of Tau to dendritic spines (*Zhao et al., 2016*). However, the physiological significance of Tau mutant mislocalization and its effect on Fyn nanoclustering is currently not known. We therefore investigated whether the P301L mutant Tau affects the nanoscale organization of Fyn in dendrites, compared to that of WT Tau, by carrying out sptPALM on Tau KO neurons co-expressing Fyn-mEos2, mCardinal and either TauWT-GFP (*Figure 6a*) or TauP301L-GFP (*Figure 6b*). We found that Fyn mobility dramatically decreased in the dendrites of Tau KO neurons expressing TauP301L-GFP compared to those expressing TauWT-GFP, as evidenced by the changes in the average MSD (*Figure 6c*), the AUC (*Figure 6d*), the frequency distribution of diffusion coefficients (*Figure 6e*), and the immobile fraction (*Figure 6f*). We then identified trajectories from spines and determined whether P301L mutant Tau altered the mobility and nanoscale organization of Fyn in these regions. We found that the mobility of Fyn decreased significantly in the spines of Tau KO neurons expressing TauP301L-GFP compared to those expressing TauWT-GFP, as assessed by the changes in the average MSD (*Figure 6g*), the AUC (*Figure 6h*), the frequency distribution of diffusion coefficients (*Figure 6i*), and the immobile fraction (*Figure 6j*). These results demonstrate a strong effect of P301L mutant Tau expression on Fyn mobility, and suggest for the first time that the complex interaction dynamics between Tau and Fyn underpin the nanoscale organization of Fyn in the dendritic shaft under physiological conditions and in spines under pathological conditions.

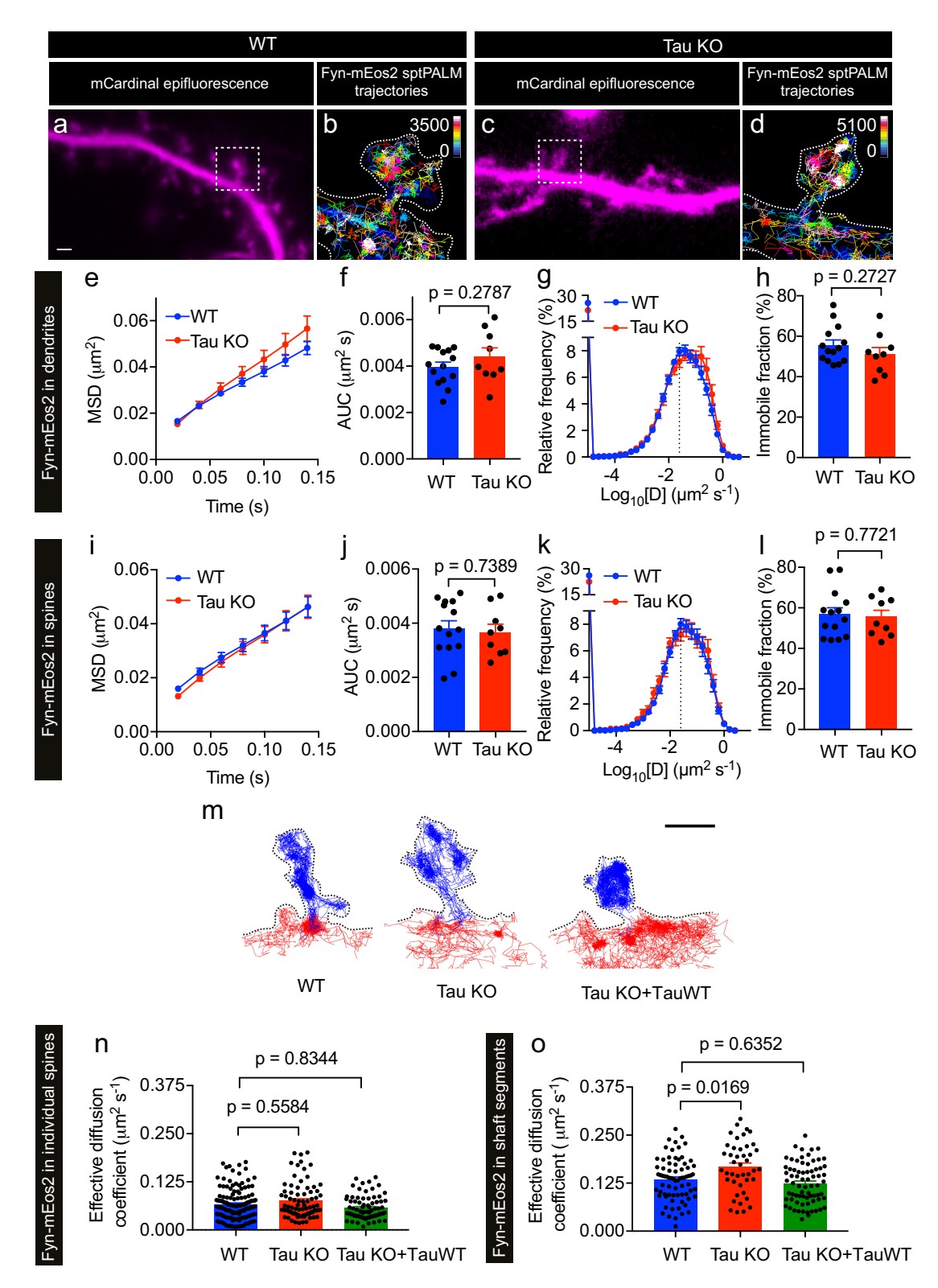

**Figure 5.** Loss of Tau alters Fyn mobility in the dendritic shafts but not in the spines. (a–d) Representative epifluorescence images of WT (a) and Tau KO (c) neurons co-expressing mCardinal and Fyn-mEos2. sptPALM trajectory maps of Fyn-mEos2 molecules in WT (b) and Tau KO (d) neurons. Regions marked in (a) and (c) are shown at higher magnification in (b) and (d), respectively. Scale bar, 1 μm. (e–l) Comparison of Fyn mobility in dendrites (e–h) and in spines (i–l) of WT (blue) or Tau KO (red) neurons. Mean ± SEM values were obtained from n = 14 WT neurons and n = 9 Tau KO neurons. (e, i)

*Figure 5 continued on next page*

*Figure 5 continued*

The average mean square displacement (MSD) as a function of time. (**f, j**) The corresponding area under the MSD curves (AUC). (**g, k**) The distribution of diffusion coefficients (D) shown in a semi-log plot. The dashed line distinguishes the immobile and mobile fractions. (**h, l**) The corresponding immobile fraction. (**m**) Examples of trajectories of Fyn-mEos2 molecules in an individual spine (blue) and the immediately adjacent shaft region (red) of WT (left), Tau KO (middle) and Tau KO expressing TauWT-GFP neurons. Scale bar, 1 μm. (**n**) Effective diffusion coefficients ($D_{eff}$) of the trajectories belonging to individual spines of WT, Tau KO and Tau KO + TauWT neurons. (**o**) $D_{eff}$ of the trajectories belonging to individual shaft regions of WT, Tau KO and Tau KO + TauWT neurons. In (**n**) and (**o**), spines or shaft segments that contained at least 50 trajectories were used to compute $D_{eff}$. Statistical comparisons were performed using Student's *t*-test (f, h, j, l) or Kruskal-Wallis test with Dunn's post hoc multiple comparisons test (n, o) . The adjusted p-values accounting for multiple comparisons are reported in (n) and (o).
DOI: https://doi.org/10.7554/eLife.45040.009

The following figure supplement is available for figure 5:

**Figure supplement 1.** Estimation of the effective diffusion coefficient of Fyn trajectories in dendritic spines and shaft segments.
DOI: https://doi.org/10.7554/eLife.45040.010

## FTD-linked P301L mutant Tau immobilizes Fyn in spines

P301L mutant Tau mislocalizes to dendritic spines and impairs spine function (*Hoover et al., 2010*). In order to further dissect the effect of P301L mutant Tau on the organization of Fyn within individual spines, we computed the average MSD and $D_{eff}$ value of trajectories belonging to individual spines of Tau KO neurons expressing either TauWT-GFP or TauP301L-GFP. This analysis revealed that the average MSD and $D_{eff}$ value were significantly lower in individual spines of Tau KO neurons expressing TauP301L-GFP than in those expressing TauWT-GFP (*Figure 7a,b*), further supporting our earlier findings (*Figure 6g–6j*). We then performed an MSS analysis of Fyn trajectories (see *Figure 3*). The $S_{MSS}$ computed from the spine trajectories of Tau KO neurons expressing TauP301L-GFP (mean $S_{MSS}$ = 0.1) was found to be significantly lower than that of Tau KO neurons expressing TauWT-GFP (mean $S_{MSS}$ = 0.13), indicating that Fyn molecules are more confined in the spines of Tau KO neurons expressing P301L mutant Tau (*Figure 7c*). We also analyzed the cumulative distribution of displacements of Fyn-mEos2 molecules at 20 ms intervals by applying the three-diffusive-state model (*Figure 7d–e*). We found that, although the apparent diffusion coefficients of each state were similar (*Figure 7d*), the state occupancy of the immobile state was not significantly different, the state occupancy of the confined state increased and that of the free state significantly decreased in spines of Tau KO neurons expressing TauP301L-GFP when compared to those expressing TauWT-GFP (*Figure 7e*). Finally, using a Voronoï-tessellation-based spatial clustering algorithm (*Levet et al., 2015*), we computed the nanodomain diameter and area, as well as the number of nanodomains per spine. The nanodomain diameter and area were not significantly different between the two conditions, but the number of nanodomains per spine increased in Tau KO neurons expressing TauP301L-GFP when compared to those expressing TauWT-GFP (*Figure 7f–h*). To test whether the effect of P301L mutant Tau depends on the increased translocation of Tau into dendritic spines, we performed sptPALM of Fyn on Tau KO neurons expressing Fyn-mEos2, GFP and either TauWT-GFP or Tau lacking the microtubule-binding domain (ΔTau-GFP) that is known to access spines (*Ittner et al., 2010*; *Cummins et al., 2019*). We found that ΔTau-GFP mislocalized to dendritic spines (*Figure 8a*) but did not alter Fyn mobility in dendrites (*Figure 8b–e*) or spines (*Figure 8f–m*). Taken together, these results provide strong evidence that P301L mutant Tau confines Fyn and alters its nanoscale organization within individual spines, demonstrating that Tau acts as a key regulator of Fyn in both physiological and pathological situations.

## Discussion

Signaling mediated through Fyn kinase is critical for synaptic plasticity, and dysregulation of Fyn activity has been implicated in both Aβ and Tau pathology (*Ittner and Götz, 2011*; *Haass and Mandelkow, 2010*). Our study provides the first conceptual framework to describe how the nanoscale organization of Fyn may be essential for synaptic plasticity, with ramifications for Tau-mediated neurodegeneration. We have revealed that Fyn is organized in nanodomains and that its nanoscale organization in dendrites is altered with neuronal maturation, with the kinase being more immobile in dendritic spines compared to shafts. Furthermore, Fyn mobility is characterized by at least three distinct motion states: an immobile, a confined, and a free diffusive state. Importantly, Tau was

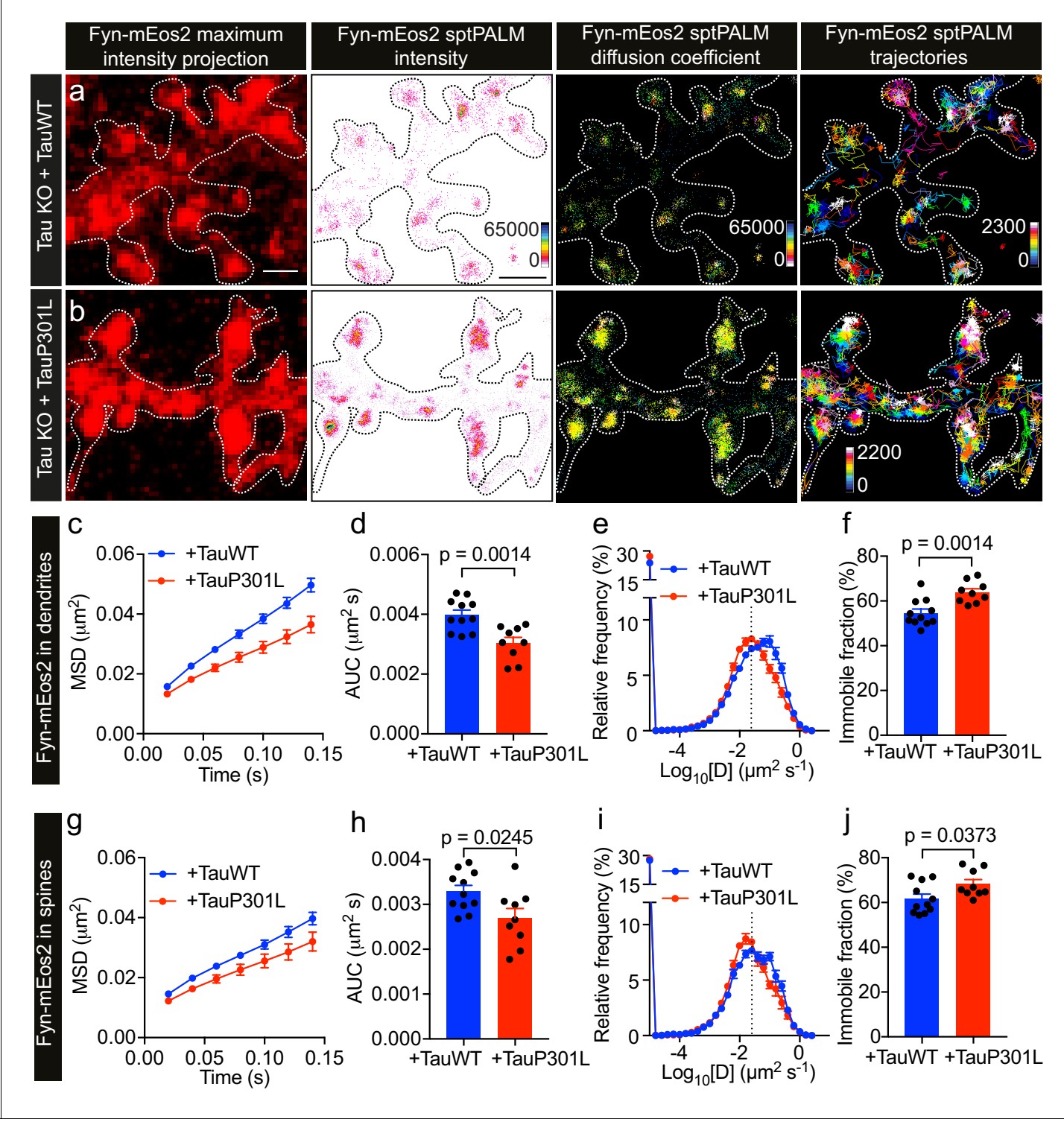

**Figure 6.** P301L mutant Tau lowers Fyn mobility in dendrites. (**a, b**) Representative low-resolution and super-resolution images of Tau KO neurons expressing mCardinal, Fyn-mEos2 and TauWT-GFP or TauP301L-GFP, obtained using sptPALM. sptPALM imaging was performed at 50 Hz for 320 s (16,000 frames) to construct the maps of localization intensities, diffusion coefficients, and trajectories of Fyn-mEos2 molecules. The cooler colors represent higher localization intensities and larger diffusion coefficients, and each trajectory is coded with a different color. (**c–j**) Comparison of Fyn mobility in the dendrites (**c–f**) and spines (**g–j**) of Tau KO neurons expressing TauWT-GFP (blue) or TauP301L-GFP. (**c, g**) The average mean square displacement (MSD) as a function of time. (**d, h**) The corresponding area under the MSD curves (AUC). (**e, i**) The distribution of diffusion coefficients (D) shown in a semi-log plot. The threshold used to distinguish the immobile ($Log_{10}[D] \leq -1.6$) and mobile ($Log_{10}[D] > -1.6$) fractions of molecules is

*Figure 6 continued on next page*

*Figure 6 continued*

indicated with a dashed line. (f, j) The corresponding immobile fraction. Mean ± SEM values were obtained from n = 11 Tau KO neurons expressing TauWT-GFP and n = 9 Tau KO neurons expressing TauP301L-GFP. Statistical comparisons were performed using Student's *t*-tests (d, f, h, and j).

DOI: https://doi.org/10.7554/eLife.45040.011

found to modulate Fyn mobility in dendritic shafts. Although the mutant Tau lacking its microtubule binding domains (ΔTau) did not affect Fyn mobility, the FTD-linked P301L mutant Tau largely increased Fyn trapping in nanodomains in the spines by increasing Fyn occupancy in the confined state and decreasing its occupancy in the free diffusive state. Mutant tau mislocalization into dendritic spines involves the formation of the truncation product ΔTau314, which lacks a part of the microtubule-binding domain as described previously (*Zhao et al., 2016*). Interestingly, tauopathy induced by low levels of truncated Tau can be rescued by pharmacological treatment (*Bondulich et al., 2016*). We have previously shown that the expression of P301L mutant Tau increases Fyn levels in PSD fractions (*Ittner et al., 2010*). Here, we show that this synaptic increase in Fyn expression drives aberrant nanoclustering in spines, potentially leading to overactivation of postsynaptic Fyn signaling.

To the best of our knowledge, our study presents the first description of the nanodomain organization of Fyn in dendrites and its regulation by Tau. Understanding the role of Fyn signaling in physiology and pathology is critical, as this kinase acts as a central hub in the integration of multiple neuronal signaling pathways that are dysregulated in disease. Fyn is targeted to dendritic spines by a process that is facilitated by the dual interactions of Tau with the SH3 domain of Fyn and of tyrosine-phosphorylated Tau with the SH2 domain of Fyn (*Ittner et al., 2010*; *Lee et al., 2004*; *Usardi et al., 2011*). Fyn may therefore regulate some of the functions of Tau in physiology and pathology. Fyn is also a critical mediator of Aβ toxicity. When overexpressed in human Aβ-forming amyloid precursor protein (APP) transgenic mouse models, Fyn accelerates synaptic and cognitive impairment (*Chin et al., 2005*). Furthermore, synaptic degeneration and memory loss are rescued when either Fyn is depleted or its activity is suppressed in an APP mutant background, such that no Aβ is produced (*Kaufman et al., 2015*; *Chin et al., 2004*). A role for Fyn in mediating the toxicity of oligomeric Aβ through the prion protein PrP$^c$ has been demonstrated in cultured neurons and in vivo (*Um et al., 2012*; *Larson et al., 2012*; *Um et al., 2013*). Fyn has further been shown to be a critical component of the ERK/rpS6 signaling cascade that facilitates Aβ-mediated increases in Tau levels and phosphorylation (*Li and Götz, 2017*). Other tyrosine kinases from the same Fyn family transduce their nanoclustering membrane organization into efficient downstream signaling during immune cell activation (*Rossy et al., 2013*). Together with the finding that the overexpression of Fyn in the brain causes an excitotoxic phenotype and early lethality (*Kojima et al., 1998*; *Xia and Götz, 2014*), our data suggest that the organization of Fyn into nanodomains may allow this kinase to orchestrate multiple signaling pathways by providing a scaffold for diverse signaling components, allowing Fyn to assume a major role in neuronal physiology and pathology. Whether Aβ mediates neurotoxicity by altering Fyn nanodomain organization is currently not known and further research would be required to assess this possibility at the nanoscale level.

The spatial spreading of activated signals from a given spine to a neighboring spine depends on the balance between the diffusion of Fyn and its deactivation rate (*Yasuda, 2017*). Trapping Fyn in nanodomains limits its diffusion, thereby restricting the kinase's signaling to individual spines. By increasing the trapping of Fyn in spines, P301L mutant Tau could potentially promote aberrant Fyn signaling, leading to neurotoxicity and subsequent neuronal degeneration. Interestingly, Fyn-overexpressing mice and patients with FTD exhibit hyperactivity (*Xia and Götz, 2014*) and repetitive compulsive behavior (*Rosso et al., 2001*), respectively. This suggests that the neurotoxicity observed in patients could be related to the excessive activity of Fyn, which is promoted by its aberrant Tau-dependent trapping in spines. By trapping Fyn in the shafts, Tau may reduce the flux of Fyn from the dendrite to the soma, which could be one potential mechanism through which Tau restricts Fyn in the dendritic compartment. The trapping of Fyn in the shaft may also limit its access to spines and could therefore fine tune the extent of Fyn clustering and signaling in this essential compartment.

The diffusion coefficients of Fyn varied by at least two orders of magnitude even in the spines of Tau KO neurons, suggesting that several other factors, such as interaction with PSD-95 (*Won et al.,*

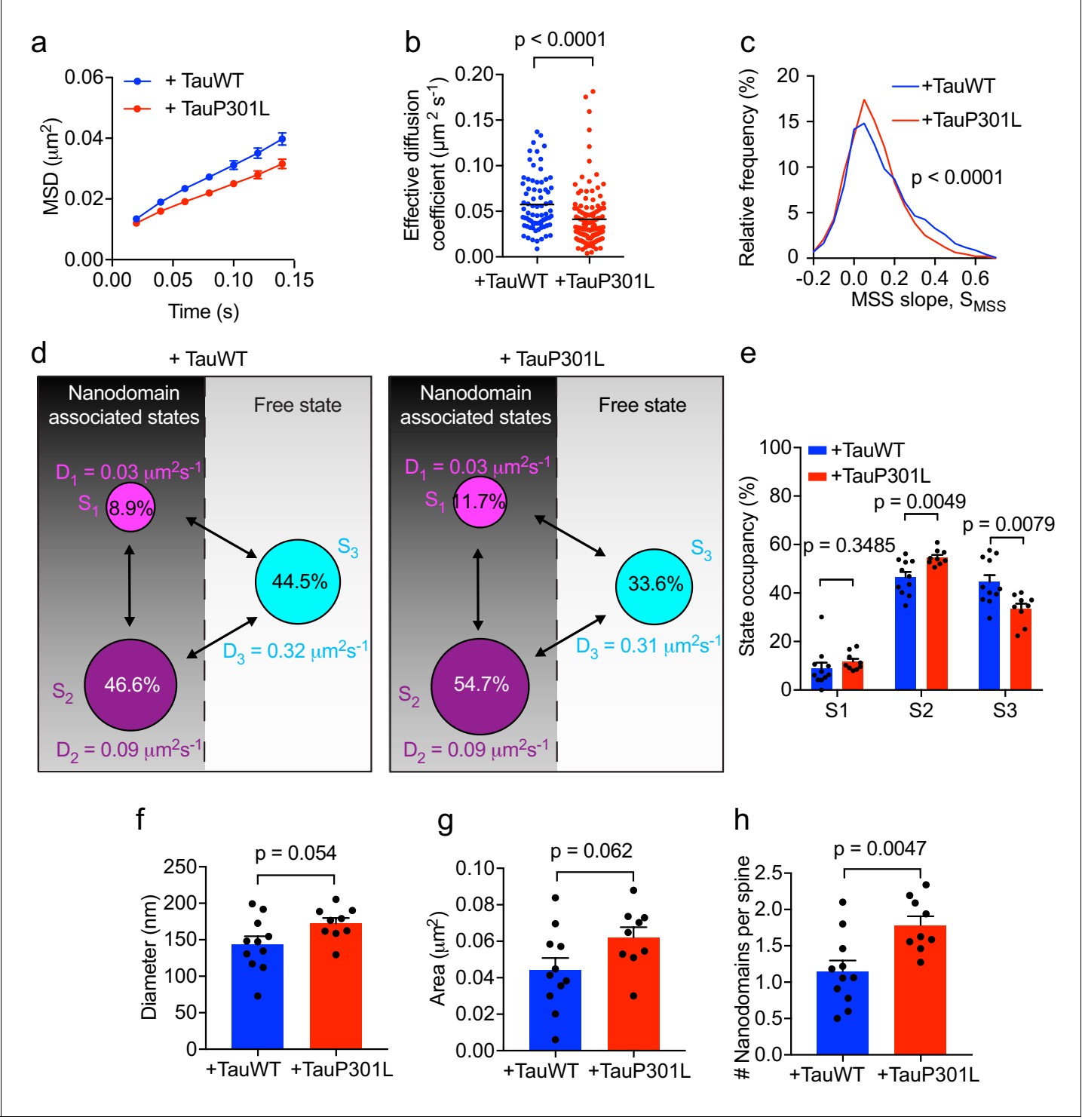

**Figure 7.** Expression of mutant Tau-P301L traps Fyn in spines. (**a**) The average mean square displacement (MSD) of Fyn-mEos2 trajectories from individual spines of Tau KO neurons expressing TauWT-GFP (n = 76 spines) or TauP301L-GFP (n = 124 spines). (**b**) The effective diffusion coefficient computed from the MSD of individual spines. (**c**) Distribution of the moment scaling spectrum slope ($S_{MSS}$) of trajectories lasting for at least 20 frames (n = 2282 trajectories from the spines of 11 Tau KO neurons expressing TauWT-GFP and n = 3764 trajectories from the spines of 9 Tau KO neurons expressing TauP301L-GFP). (**d**) A three-state model showing the inferred apparent diffusion coefficients and state occupancies (state S1, immobile; state S2, confined; state S3, apparently free), with the circle area being proportional to the state occupancy. (**e**) Comparison of the estimated state occupancies. (**f, g**) Comparison of average nanodomain diameter (**f**) and area (**g**). (**h**) Comparison of the number of nanodomains per spine. Statistical comparisons were performed using the Kruskal-Wallis test (**b**), the Kolmogorov-Smirnov test (**c**) or Student's t-test (**e–h**).

DOI: https://doi.org/10.7554/eLife.45040.012

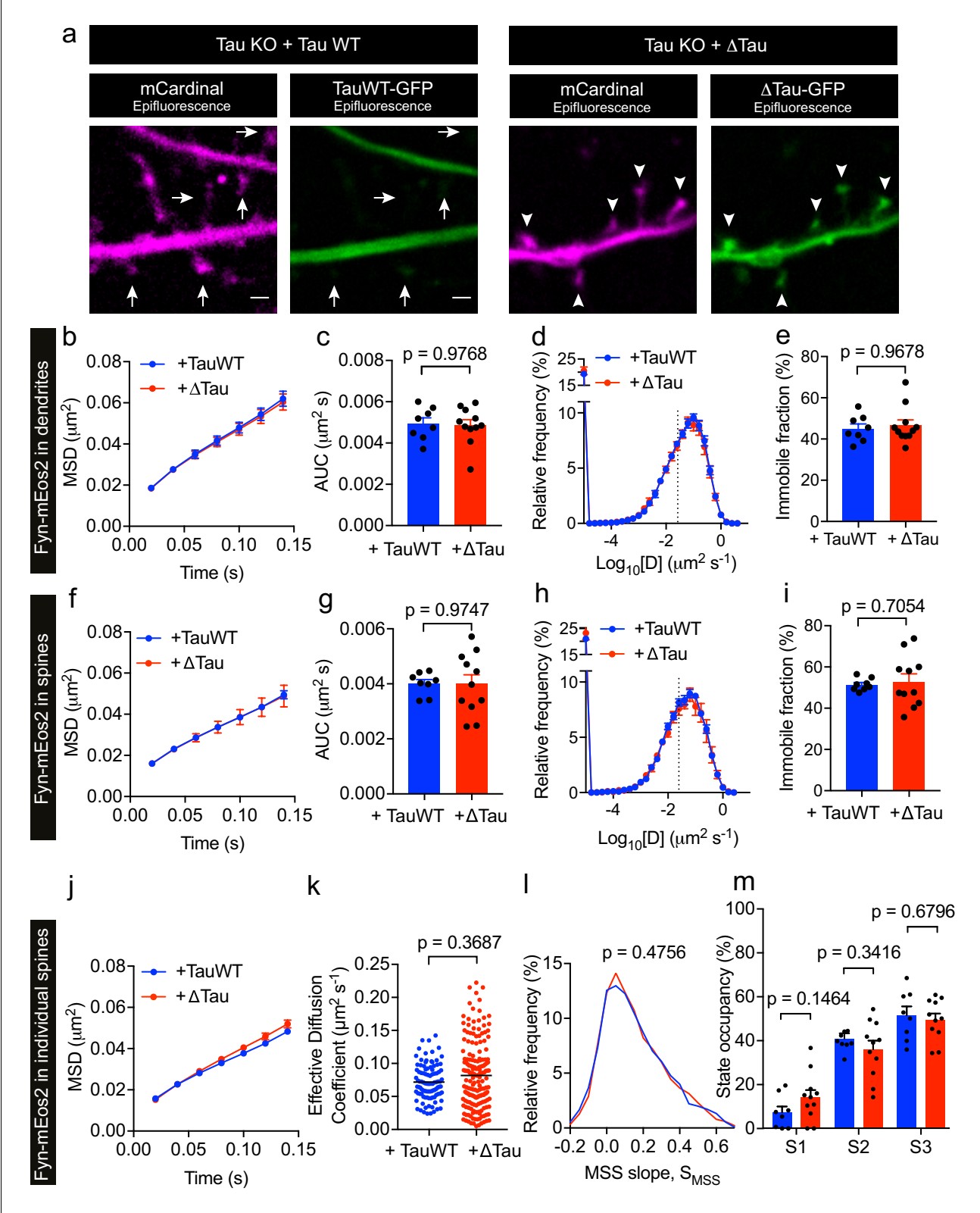

**Figure 8.** Tau lacking the microtubule-binding domain (ΔTau) mislocalizes to dendritic spines but does not affect Fyn mobility in dendrites and spines. (a) Representative epifluorescence images of Tau KO neurons co-expressing either full-length Tau (TauWT-GFP) and mCardinal or Tau lacking the microtubule-binding domain (ΔTau-GFP) and mCardinal. Arrows denote reduced localization of TauWT-GFP in dendritic spines and arrowheads denote increased mislocalization of ΔTau-GFP in dendritic spines. Scale bar, 1 μm. (b–i) Comparison of Fyn mobility in the dendrites (b–e) and spines

*Figure 8 continued on next page*

Figure 8 continued

(f–i) of Tau KO neurons expressing Fyn-mEos2, TauWT-GFP and GFP (blue) and Tau KO neurons expressing Fyn-mEos2, ΔTau-GFP and GFP. (b, f) The average mean square displacement (MSD) as a function of time. (c, g) The corresponding area under the MSD curves (AUC). (d, h) The distribution of diffusion coefficients (D) shown in a semi-log plot. (e i,) The corresponding immobile fraction. (j) The average mean square displacement (MSD) of Fyn-mEos2 trajectories from individual spines of Tau KO neurons expressing TauWT-GFP (n = 99 spines) or ΔTau-GFP (n = 181 spines). (k) The effective diffusion coefficient computed from the MSD of individual spines. (l) The distribution of the moment scaling spectrum slope ($S_{MSS}$) of trajectories lasting for at least 20 frames (n = 2561 trajectories from the spines of 8 Tau KO neurons expressing TauWT-GFP and n = 4943 trajectories from the spines of 11 Tau KO neurons expressing ΔTau-GFP). (m) The comparison of the state occupancies estimated using a three-state model. In (b–i) and (m), mean ± SEM values were obtained from n = 8 Tau KO neurons expressing Fyn-mEos2, TauWT-GFP and GFP and n = 11 Tau KO neurons expressing Fyn- mEos2, ΔTau-GFP and GFP. Statistical comparisons were performed using the Mann-Whitney U test (c, e and k), Student's t-test with Welch's correction (g, i), the Kolmogorov-Smirnov test (l) and Student's t-test (m).

DOI: https://doi.org/10.7554/eLife.45040.013

2016), molecular crowding (*Li et al., 2016*) and spine geometry (*Byrne et al., 2011*), regulate the trapping and nanodomain organization of Fyn. Recent studies have shown that P301L mutant Tau translocates to dendritic spines (*Xia et al., 2015*; *Hoover et al., 2010*), has an increased binding affinity towards Fyn in vitro (*Bhaskar et al., 2005*), and promotes actin polymerization in the presynaptic nerve terminal (*Zhou et al., 2017*). These factors could potentially contribute to the observed Tau P301L mutant-dependent increased trapping of Fyn in nanodomains in dendritic spines.

For simplicity, we interpreted Fyn mobility in spines with a three-diffusive-state model. The immobile state with the lowest diffusion coefficient is likely to represent Fyn molecules bound to their substrate, the confined state could represent Fyn molecules that are trapped in the nanodomains, and the free state may represent Fyn molecules that are moving between synaptic and extrasynaptic compartments. Interestingly, a recent study reported that CaMKII also displays at least three motion states within spines, and that NMDA receptor stimulation immobilizes CaMKII in spines (*Lu et al., 2014*). It therefore appears that the mobility patterns of signaling molecules such as CaMKII and Fyn could represent the activational state of the underlying signaling network. Investigating synaptic-activity-dependent changes in the mobility patterns of key signaling molecules could potentially unravel the biochemical signaling principles underlying synaptic function.

We found that Fyn forms nanodomains of ~170 nm in diameter within dendritic spines, which are similar to those described for PSD-95 (*Nair et al., 2013*). Fyn and PSD-95 interact on the plasma membrane following the palmitoylation of specific residues (*Sato et al., 2009*; *Tezuka et al., 1999*; *Topinka and Bredt, 1998*), and act as a hub for multiple other proteins in dendritic spines. Fyn nanoclusters resemble those formed by PSD-95, which are larger than those formed by AMPA receptors (*Nair et al., 2013*). Further, we found that Fyn forms, on average, one nanodomain per spine, which is similar to the frequency of nanodomains formed by PSD-95 (*Hruska et al., 2018*). As PSD-95 nanoclusters are remodeled during synaptic plasticity (*Hruska et al., 2018*), it would be interesting in the future to assess whether PSD-95 and Fyn are part of the same synaptic complex regulated during synaptic plasticity.

Other factors could potentially influence Fyn nanoclustering in dendrites. A recent study found that, by recruiting GluA1-containing AMPA receptors (AMPARs), netrin-1 regulates spine maturation in hippocampal neurons (*Glasgow et al., 2018*). Given that netrin-1 activates Fyn to regulate axon growth and guidance (*Meriane et al., 2004*), it is possible that netrin-1 also contributes to spine maturation via Fyn clustering and signaling. Similarly, Semaphorin3A promotes the maturation of spines in cortical neurons through Fyn signaling (*Morita et al., 2006*) and could also influence Fyn nanoclustering during maturation. Whether extracellular cues such as netrin-1 and Semaphorin3A mediate spine maturation through Fyn nanoscale reorganization remains to be determined. Previous cell culture studies have shown that the mobility of other postsynaptic proteins, such as the AMPARs (*Hoze et al., 2012*) and Neuroligin-1 (*Chamma et al., 2016*), decreases and that the NMDAR nanodomain area changes with neuronal maturation (*Kellermayer et al., 2018*). The nanoscale reorganization of postsynaptic proteins therefore appears to be a common feature associated with spine maturation, which could stem from a number of factors pertaining to the spatial trapping of key molecules that are essential for synapse formation and function. Given that the maturation of hippocampal neurons from Tau KO mice is delayed in cell culture (*Dawson et al., 2001*), it would be

interesting to determine how Tau and Fyn interact to generate the appropriate signals that are required for dendritic spine formation and maturation (*Morita et al., 2006*).

Although our results reveal that dendritic spines exhibit significantly lower Fyn mobility than dendritic shafts, we were able to detect Fyn nanodomains in shafts. Recently, the presence of gephyrin clusters in dendritic shafts was found to affect the stability of nearby spines (*Isshiki et al., 2014*). It is therefore tempting to speculate that Fyn nanodomains may have a similar functional role in this compartment. One possibility is that these nanodomains contribute to the generation of new spines (*Morita et al., 2006*). Alternatively, they may act as an extrasynaptic reservoir of Fyn. We also detected Fyn nanodomains at the base of dendritic spines, and a recent study found that AMPAR nanodomains at the base of spines can prevent the movement of AMPARs from the shaft to the spines (*Hoze and Holcman, 2014*). It is conceivable that similar Fyn nanodomains at the base of spines regulate the exchange of Fyn between synaptic and extrasynaptic regions. Alternatively, these Fyn nanodomains may assume other signaling roles or could represent a remnant of cytoskeletal remodeling at this site.

The fact that Tau interacts with Fyn (*Lee et al., 1998*; *Bhaskar et al., 2005*) could underpin some of the nanoclustering properties of Fyn, but how Tau affects Fyn organization in the dendritic shaft is currently not understood. Tau has been shown to bind to microtubules, and its dynamic properties have been elegantly described at a single-molecule level (*Janning et al., 2014*); however, whether Tau itself is organized in nanodomains is currently unknown. Further work is therefore needed to understand the nanoscale organization of Tau and its physiological and pathological implications.

In summary, our study demonstrates that Fyn is organized into specific nanodomains that are dynamically controlled by Tau in dendrites. Tau plays a pivotal role in controlling the nanoscale organization of Fyn molecules. Importantly, mutations of Tau that are associated with the development of frontotemporal dementia have major effects, inducing aberrant trapping of Fyn in spines, which may contribute to the development of the pathology. Fyn is therefore a putative new target for the pathological effects elicited by Tau FTD mutations.

# Materials and methods

## Key resources table

| Reagent type (species) or resource | Designation | Source or reference | Identifiers | Additional information |
|---|---|---|---|---|
| Strain, strain background (*M. musculus*) | C57Bl/6 | Jackson Laboratory | Cat. #000664 | NA |
| Genetic reagent (*M. musculus*) | Tau KO (*Mapt$^{-/-}$*) | Jackson Laboratory (*Dawson et al., 2001*) | Cat. #007251 | Prof. Michael Vitek (Duke University Medical Center) |
| Antibody | Rabbit anti-Fyn | Cell Signalling Technologies | Cat. #4023 | IF (1:500) |
| Antibody | Mouse anti-PSD-95 | Merck Millipore | Cat. #MABN68 | IF (1:500) |
| Antibody | Chicken anti MAP2 | Merck Millipore | Cat. #AB15452 | IF (1:500) |
| Recombinant DNA reagent | mCardinal-N1 | Addgene | Cat. #54590 | |
| Recombinant DNA reagent | pEGFP-N1 | Clontech/NovoPro | Cat. #V12021 | |
| Recombinant DNA reagent | Fyn-mEos2 | This paper | | |
| Recombinant DNA reagent | Tau-GFP | PMID: 27378256 | | Prof. Jürgen Götz (Queensland Brain Institute, University of Queensland) |
| Recombinant DNA reagent | Tau-P301L-GFP | PMID: 27378256 | | Prof. Jürgen Götz (Queensland Brain Institute, University of Queensland) |

*Continued on next page*

*Continued*

| Reagent type (species) or resource | Designation | Source or reference | Identifiers | Additional information |
|---|---|---|---|---|
| Recombinant DNA reagent | ΔTau-GFP | *Cummins et al., 2019* | | Prof. Jürgen Götz (Queensland Brain Institute, University of Queensland) |
| Chemical compound, drug | Phalloidin-Alexa Fluor 647 | NEB | Cat. #8940S | IF (1:100) |
| Software, algorithm | Fiji-ImageJ | ImageJ (http://imagej.nih.gov/ij/) | RRID:SCR_003070 | Version 2.0.0-rc-68/1.52e |
| Software, algorithm | Neurolucida | MBF Bioscience | RRID:SCR_001775 | |
| Software, algorithm | Huygens software | Scientific Volume Imaging | RRID:SCR_014237 | |
| Software, algorithm | Metamorph software | Molecular Devices (https://www.moleculardevices.com) | RRID:SCR_002368 | Version 7.7.8 |
| Software, algorithm | PALMTracer | http://www.iins.u-bordeaux.fr/team-sibarita-PALMTracer | | |
| Software, algorithm | SR-Tesseler | *Levet et al., 2015* (http://www.iins.u-bordeaux.fr/team-sibarita-SR-Tesseler) | | |
| Software, algorithm | SharpViSu | *Andronov et al., 2016* (https://github.com/andronovl/SharpViSu) | | |
| Software, algorithm | DC-MSS | *Vega et al., 2018* (https://github.com/kjaqaman/DC-MSS) | | |
| Software, algorithm | Graphpad Prism | GraphPad Prism (https://graphpad.com) | RRID:SCR_015807 | Version 7.0d |

## Animal ethics and mouse strains

All experimental procedures were conducted under the guidelines of the Australian Code of Practice for the Care and Use of Animals for Scientific purposes and were approved by the University of Queensland Animal Ethics Committee (QBI/412/14/NHMRC; QBI/027/12/NHMRC; QBI/254/16/NHMRC). Mice were maintained in a 12-hr light/dark cycle and housed in a PC2 facility with *ad libitum* access to food and water. Wildtype mice (C57Bl/6 strain) and Tau KO (*Mapt*$^{-/-}$) mice on a C57Bl/6 background (*Dawson et al., 2001*) were used throughout the study. Tau KO has a loss of function of the *Mapt* gene.

## Primary hippocampal cultures

Embryonic day (E)16 hippocampal neurons were obtained from wildtype and Tau KO mice (*Dawson et al., 2001*) and prepared as described previously (*Joensuu et al., 2017*). Briefly, for live-cell super-resolution microscopy, 100,000 neurons were plated onto poly-L-lysine-coated 35 mm glass-bottom dishes (In Vitro Scientific). For immunocytochemistry, 80,000 cells/well were plated onto poly-L-lysine-coated 10 mm diameter coverslips (ProSciTech) in a 12-well plate (*Fath et al., 2009*). The neurons were cultured in Neurobasal medium (Gibco) supplemented with 5% fetal bovine serum (Hyclone), 2 mM Glutamax (Gibco) and 50 U/mL penicillin/streptomycin (Invitrogen). The medium was changed to serum-free Neurobasal medium supplemented with 2% B27 (Gibco) 4 hr post-seeding, and half the medium was changed every week.

## Immunocytochemistry

Primary hippocampal neurons were fixed with 4% paraformaldehyde/4% sucrose for 15 min at room temperature, permeabilized with 0.2% Triton X-100 for 10 min, then blocked for 1 hr in 5% goat

serum, followed by primary antibody incubation overnight at 4°C and secondary antibody incubation for 1 hr at room temperature. The following primary antibodies were used: Fyn (Cell Signalling Technologies #4023; rabbit polyclonal, 1:500), PSD-95 (Millipore, monoclonal, 1:500), MAP2 (Millipore, chicken polyclonal, 1:500), and phalloidin-Alexa Fluor 647 (NEB, 1:100), which was used to detect actin. As secondary antibodies, we used Alexa-Fluor-488-labeled goat-anti-rabbit antibody, Alexa-Fluor-555-labeled goat anti-mouse antibody and Alexa-Fluor-647-labeled goat anti-chicken antibody (all from LifeTechnologies, Thermo Fisher, 1:500). Fluorescence images were captured with a 20X or a 60X objective on a Zeiss LSM710 confocal microscope and analyzed with Fiji-ImageJ software (*Schindelin et al., 2012*).

## Cloning

Fyn-mEos2 (human Fyn with carboxy-terminal mEos2 tag) was subcloned from a Fyn-pGEM-T Easy plasmid (human full-length Fyn, isoform 1) by ApaI/SalI double digestion and ligated into the pEGFP-N1 (Clontech) and mEos2-N1 (Addgene #54662) vector.

## Super-resolution microscopy with oblique illumination

Primary neurons were transfected using Lipo2000 and used for super-resolution microscopy 5–7 days post-transfection. The following constructs were used for transfection: Fyn-mEos2, mCardinal-N1 (Addgene #54590), Tau-EGFP (human Tau with carboxy-terminal EGFP tag) (*Xia et al., 2015*), ΔTau-EGFP (human Tau lacking the last 186 amino acids, with a carboxy-terminal EGFP tag) and Tau-P301L-EGFP (human mutated Tau-P301L with a carboxy-terminal EGFP tag) (*Xia et al., 2015*). For live-cell super-resolution microscopy with oblique illumination, Fyn-mEos2-transfected neurons were bathed in imaging buffer (145 mM NaCl, 5.6 mM KCl, 2.2 mM $CaCl_2$, 0.5 mM $MgCl_2$, 5.6 mM D-glucose, 0.5 mM ascorbic acid, 0.1% BSA, 15 mM HEPES, pH 7.4). Neurons were visualized at 37° C on a Roper Scientific TIRF microscope equipped with an ILas[2] double laser illuminator (Roper Scientific), a Nikon CFI Apo TIRF 100×/1.49 N.A. objective (Nikon Instrument), an Evolve512 delta EMCCD camera (Photometrics) and a perfect focus system, allowing acquisitions in oblique illumination. Image acquisition was performed using Metamorph software (version 7.7.8, Molecular Devices).

## 3D analysis of sptPALM trajectories

The multi-dimensional acquisition Z-series App from Metamorph software was used to obtain a z-stack acquisition (each frame every 0.1 µm) of the epifluorescence signal corresponding to the pEGFP fluorophore of hippocampal neurons (DIV20-22) expressing Fyn-mEos2 and pEGFP. sptPALM imaging of the same neuronal segment was performed immediately after the stack acquisition. Z-stack images were deconvolved using Huygens software (Scientific Volume Imaging). 2D maximum intensity projection was performed using Fiji-ImageJ software (*Schindelin et al., 2012*). 3D analysis of the dendritic architecture was performed using Neurolucida software (MBF Bioscience). Only spines protruding from the sides of the dendrite branch were considered in this analysis. The shaft region was selected by excluding shaft segments containing spines projecting away from the imaging plane. sptPALM trajectories from spines or shafts were selected for further analysis.

## Single-particle tracking photoactivated localization microscopy (sptPALM)

Time-lapse movies were acquired at 50 Hz (16,000 frames) at 37°C. For sptPALM, a 405-nm laser was used to photo-activate the cells expressing Fyn-mEos2 and a 561-nm laser was used simultaneously for excitation of the resulting photo-converted single molecules. To isolate the mEos2 signal from auto-fluorescence and background signals, a double-beam splitter (LF488/561-A-000, Semrock) and a double-band emitter (FF01-523/610-25, Semrock) were used. To spatially distinguish and temporally separate the stochastically activated molecules during acquisition, the respective power of the lasers was adjusted. The 405-nm laser was used between 1.5% and 3% of the initial laser power (100 mW Vortran Laser Technology), and the 561-nm laser was used at 70% of the initial laser power (150 mW Cobolt Jive).

## Single-particle trajectory analysis

The localization and tracking of single molecules were performed as previously described (*Nair et al., 2013*). Briefly, single-molecule localizations were detected using a wavelet-based segmentation, and trajectories were computed using a simulated annealing-based tracking algorithm (*Izeddin et al., 2012*) with PALM-Tracer, software that operates with Metamorph (Molecular Devices) (*Nair et al., 2013*; *Kechkar et al., 2013*). Trajectories that lasted at least eight frames were reconstructed and the mean square displacement (MSD) was computed for each trajectory. Cells with at least 1000 trajectories were considered for further analysis. The MSD was fitted by the equation $\mathrm{MSD(t)} = \mathrm{a} + 4\mathrm{Dt}$, where D is the diffusion coefficient, a is the y-intercept and t is the time. We considered trajectories with $\mathrm{Log}_{10}[D] \leq -1.6$ as immobile (*Bademosi et al., 2017*; *Constals et al., 2015*; *Kasula et al., 2016*) and computed the immobile fraction from the distribution of the diffusion coefficient histograms for statistical comparisons. The moment scaling spectrum (MSS) analysis was applied to Fyn trajectories that lasted for at least 20 frames, as described previously (*Vega et al., 2018*).

## SR-Tesseler analysis

Fyn nanodomains were quantified from sptPALM data using SR-Tesseler analysis (*Levet et al., 2015*). Briefly, the coordinates of single-molecule localizations were used to construct a Voronoï diagram, which segmented the sptPALM data into polygons centered on individual localizations. Object segmentation then provided the neuronal contour. We then identified potential Fyn nanodomains as regions within the neuronal contour that contained at least 50 detections and had a local density at least two-fold greater than the average density of the identified object. The nanodomain diameter was calculated using principal component analysis and the area was determined from the segmented cluster outline, as described previously (*Levet et al., 2015*). A cross-correlation-based drift correction of the sptPALM data was performed using the SharpViSu tool (*Andronov et al., 2016*) before performing SR-Tesseler analysis.

## Step-length analysis

We estimated the diffusion coefficients and the occupancies of multiple kinetic states by analyzing the cumulative distribution function (CDF) of displacements of Fyn-mEos2 molecules at 20 ms intervals. To avoid bias resulting from long trajectories (*Chen et al., 2015*), we computed the CDF of each cell by including only the first seven displacements of each trajectory, which is the minimum trajectory length considered in all our of analyses. The CDF was then fitted with a three-state model given by:

$$C(r,t) = 1 - f_1 exp\left(\frac{-r^2}{4D_1 t}\right) - f_2 exp\left(\frac{-r^2}{4D_2 t}\right) - f_3 exp\left(\frac{-r^2}{4D_3 t}\right) \tag{1}$$

Here, r is the displacement, $\Delta t$ is the time interval (20 ms), $D_1$, $D_2$ and $D_3$ are the diffusion coefficients of the three states, and $f_1$, $f_2$ and $f_3$ are the state occupancies. We computed the empirical CDF using the tool ECDF and fit the model predictions to the data using the non-linear regression tool NLINFIT in Matlab to estimate the model parameters. We performed global fitting for each condition by keeping the diffusion coefficients of each state constant across cells and allowing the state occupancies to vary across cells. This was necessary to estimate the parameters reliably.

## Statistics

The D'Agostino and Pearson test was used to test for normality. The Student's *t*-test was used when the data were normally distributed, and the non-parametric Mann Whitney U test was used when the data were not normally distributed. For data sets that compared more than two groups, an ANOVA was used with corrections for multiple comparisons. Statistical comparisons were performed on a per-cell basis, with neurons collected from at least three independent transfection experiments. Unless otherwise stated, values are represented as the mean ± SEM. The tests used are indicated in the respective figure legends. Data were considered significant at $p < 0.05$. Statistical tests were performed and figures were made using GraphPad Prism 7. A summary of statistical analyses is provided in *Supplementary file 1*.

## Acknowledgements

The authors thank Rowan Tweedale for the critical appraisal of the manuscript, Rumelo Amor and his team at the Queensland Brain Institute (QBI) for their excellent support with the microscopy, Jake Carroll and his team at the QBI for providing computing infrastructure support, Dr. Vanessa Lanoue for her help in the deconvolution of the images, and Jean-Baptiste Sibarita (IINS, CNRS/University of Bordeaux) for his kind support and help with the single-molecule analysis software PALMtracer. PP was supported by a University of Queensland Postdoctoral Fellowship. This work was supported by the Federal Government of Australia (ACT900116) and the State Government of Queensland (DSITI, Department of Science, Information Technology and Innovation), by the National Health and Medical Research Council of Australia (GNT1145580 to JG and GNT1127999 to JG and FAM), and by an Australian Research Council linkage infrastructure equipment and facilities grant (LE130100078 to FAM) and a NHMRC Senior Research Fellowship (GNT1060075 to FAM).

## Additional information

### Funding

| Funder | Grant reference number | Author |
| --- | --- | --- |
| National Health and Medical Research Council | GNT1127999 | Jürgen Götz<br>Frédéric A. Meunier |
| National Health and Medical Research Council | GNT1145580 | Jürgen Götz |
| National Health and Medical Research Council | GNT1060075 | Frédéric A. Meunier |
| Federal Government of Australia | ACT900116 | Jürgen Götz |
| Australian Research Council | LE130100078 | Frédéric A. Meunier |
| University of Queensland | Postdoctoral fellowship | Pranesh Padmanabhan |

The funders had no role in study design, data collection and interpretation, or the decision to submit the work for publication.

### Author contributions

Pranesh Padmanabhan, Ramón Martínez-Mármol, Conceptualization, Data curation, Software, Formal analysis, Validation, Investigation, Methodology, Writing—original draft, Writing—review and editing; Di Xia, Conceptualization, Investigation; Jürgen Götz, Conceptualization, Supervision, Funding acquisition, Investigation, Methodology, Writing—original draft, Project administration, Writing—review and editing; Frédéric A Meunier, Conceptualization, Data curation, Supervision, Funding acquisition, Investigation, Methodology, Writing—original draft, Project administration, Writing—review and editing

### Author ORCIDs

Pranesh Padmanabhan (iD) https://orcid.org/0000-0001-5569-8731
Jürgen Götz (iD) https://orcid.org/0000-0001-8501-7896
Frédéric A Meunier (iD) https://orcid.org/0000-0001-6400-1107

### Ethics

Animal experimentation: All experimental procedures were conducted under the guidelines of the Australian Code of Practice for the Care and Use of Animals for Scientific purposes and were approved by the University of Queensland Animal Ethics Committee (QBI/412/14/NHMRC; QBI/027/12/NHMRC; QBI/254/16/NHMRC).

### Decision letter and Author response

Decision letter https://doi.org/10.7554/eLife.45040.017

Author response https://doi.org/10.7554/eLife.45040.018

## Additional files

### Supplementary files

• Supplementary file 1. Summary of statistical analyses.
DOI: https://doi.org/10.7554/eLife.45040.014

• Transparent reporting form
DOI: https://doi.org/10.7554/eLife.45040.015

### Data availability

All data generated or analysed in this study are within the paper and supplementary materials.

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
