## [Decision Letter]

Thank you for submitting your article "Frontotemporal dementia mutant Tau promotes aberrant Fyn nanoclustering in dendritic spines" for consideration by *eLife*. Your article has been reviewed by three peer reviewers, one of whom is a member of our Board of Reviewing Editors, and the evaluation overseen by Huda Zoghbi as the Senior Editor. The following individual involved in review of your submission has agreed to reveal their identity: Dezhi Liao (Reviewer #2).

The reviewers discussed the reviews with one another and the Reviewing Editor drafted this decision to help you prepare a revised submission.

Summary:

Using cultured hippocampal neurons and an elegant adaptation of super-resolution microscopy along with image analyses (sptPALM), this study reveals that Fyn in dendrites forms discrete dynamic nanoscale domains whose mobility/localization alters with neuronal development. Results in this well done study are solid and novel in that they are the first sptPALM study of Fyn dynamics in dendrites.

Essential revisions:

Tau proteins lacking the P301L mutation that are mis-sorted to spines (i.e., truncated tau) need to be tested, see Zhao et al., 2016. A truncated tau experiment would provide more information (the role of microtubule/actin binding). In addition, since Fyn did not need tau to be clustered in spines, it is puzzling why tau P301L would have an effect in spines. There was not enough discussion of this result, which was presented in 2 out of 7 figures.

Finally, data comparing the level of Fyn-GFP in WT vs. tau^-/-^ mice in dendrites would need to be controlled by data showing comparable levels of Fyn-GFP expressed in both neurons/cultures. If such data are not available, Figure 4D, E should be deleted.

[Editors' note: further revisions were requested prior to acceptance, as described below.]

Thank you for resubmitting your work entitled "Frontotemporal dementia mutant Tau promotes aberrant Fyn nanoclustering in hippocampal dendritic spines" for further consideration at *eLife*. Your revised article has been favorably evaluated by Huda Zoghbi as the Senior Editor and a Reviewing Editor.

The manuscript has been improved but there is a remaining issue that needs to be addressed before acceptance, as outlined below:

Given the importance of the newly added ΔTau data, the figure in which these data are presented should be moved from supplementary data to a figure in the main text.

---

## [Author Response]

Essential revisions:Tau proteins lacking the P301L mutation that are mis-sorted to spines (i.e., truncated tau) need to be tested, see Zhao et al., 2016. A truncated tau experiment would provide more information (the role of microtubule/actin binding). In addition, since Fyn did not need tau to be clustered in spines, it is puzzling why tau P301L would have an effect in spines. There was not enough discussion of this result, which was presented in 2 out of 7 figures.

We appreciate this suggestion and have now performed new experiments with Tau lacking microtubule-binding domain (ΔTau; Cummins et al., 2019) to address how Tau mislocalization to dendritic spines affects the mobility and nanoscale organization of Fyn. We first observed in Tau^(−/−)^ neurons expressing ΔTau-GFP that ΔTau mislocalizes to dendritic spines (Figure 7—figure supplement 1A). Interestingly, we found that the mobility of Fyn-mEos2.1 in the dendrites and spines of Tau^(−/−)^ neurons expressing ΔTau-GFP was similar to those expressing TauWT-GFP (Figure 7—figure supplement 1B-M), suggesting that other mechanisms in addition to the mislocalization of TauP301L to spines are involved in TauP301L-mediated changes in the mobility and organization of Fyn in the dendrites and spines. As suggested, in the fourth paragraph of the Discussion, we discuss the potential factors contributing to P301L mutant Tau-dependent increase of Fyn trapping in spines. We discuss our data in the context of that of Zhao and colleagues (Zhao et al., 2016).

Finally, data comparing the level of Fyn-GFP in WT vs. tau^-/-^ mice in dendrites would need to be controlled by data showing comparable levels of Fyn-GFP expressed in both neurons/cultures. If such data are not available, Figure 4D, E should be deleted.

Following the reviewer’s suggestion, we have now removed Figure 4D, E and the relevant text.

[Editors' note: further revisions were requested prior to acceptance, as described below.]

The manuscript has been improved but there is a remaining issue that needs to be addressed before acceptance, as outlined below:Given the importance of the newly added ΔTau data, the figure in which these data are presented should be moved from supplementary data to a figure in the main text.

As requested, we have moved the figure describing the effect of ΔTau from the supplementary (Figure 7—figure supplement 1 in the previous version) to a figure in the main text (Figure 8 in the revised version). We once again thank the editors and reviewers for their constructive comments, which helped improve our manuscript.